# Feasibility of Wave Intensity Analysis from 4D Cardiovascular Magnetic Resonance Imaging Data

**DOI:** 10.3390/bioengineering10060662

**Published:** 2023-05-31

**Authors:** Froso Sophocleous, Kiril Delchev, Estefania De Garate, Mark C. K. Hamilton, Massimo Caputo, Chiara Bucciarelli-Ducci, Giovanni Biglino

**Affiliations:** 1Bristol Heart Institute, Bristol Medical School, University of Bristol, Bristol BS8 1QU, UK; 2University Hospitals Bristol and Weston NHS Foundation Trust, Bristol BS1 3NU, UK; 3Royal Brompton and Harefield Hospitals, Guys and St Thomas NHS Trust, London UB9 6JH, UK; 4School of Biomedical Engineering and Imaging Sciences, Faculty of Life Sciences and Medicine, Kings College London, London WC2R 2LS, UK; 5National Heart and Lung Institute, Imperial College London, London SW7 2BX, UK

**Keywords:** aorta, bicuspid aortic valve, magnetic resonance imaging, aortic valve, wave intensity analysis, hemodynamics, aortic distensibility, wave reflection, arterial waves

## Abstract

Congenital heart defects (CHD) introduce haemodynamic changes; e.g., bicuspid aortic valve (BAV) presents a turbulent helical flow, which activates aortic pathological processes. Flow quantification is crucial for diagnostics and to plan corrective strategies. Multiple imaging modalities exist, with phase contrast magnetic resonance imaging (PC-MRI) being the current gold standard; however, multiple predetermined site measurements may be required, while 4D MRI allows for measurements of area (A) and velocity (U) in all spatial dimensions, acquiring a single volume and enabling a retrospective analysis at multiple locations. We assessed the feasibility of gathering hemodynamic insight into aortic hemodynamics by means of wave intensity analysis (WIA) derived from 4D MRI. Data were collected in *n* = 12 BAV patients and *n* = 7 healthy controls. Following data acquisition, WIA was successfully derived at three planes (ascending, thoracic and descending aorta) in all cases. The values of wave speed were physiological and, while the small sample limited any clinical interpretation of the results, the study shows the possibility of studying wave travel and wave reflection based on 4D MRI. Below, we demonstrate for the first time the feasibility of deriving wave intensity analysis from 4D flow data and open the door to research applications in different cardiovascular scenarios.

## 1. Introduction

An estimated 1.35 million babies are diagnosed each year globally with congenital heart disease (CHD) [1]. The presence of different defects affects blood flow. Haemodynamic changes are observed, for example, in patients with valvular conditions such as bicuspid aortic valve (BAV), with deranged aortic blood flow characterised by a helical pattern [2]. This exposes the aortic valve and ascending aorta to abnormal stress, which is considered one of the drivers of aortopathy in these patients [3]. Hence, flow assessment and quantification in patients with CHD can not only provide pathophysiological insights but also potentially inform treatment. Cross-sectional imaging is an invaluable tool to perform such hemodynamic assessment [4,5]. Cardiac magnetic resonance imaging (MRI), in particular, is a non-invasive, non-ionising and sensitive tool for both the diagnosis and follow-up of CHD patients, allowing for both anatomical and functional evaluation with high reproducibility and the ability of characterising tissues with different sequences [6]. In CHD, flow measurements are required at multiple sites [7]. The development of 4D MRI has allowed time-resolved 3D velocity phase-contrast (PC) imaging for evaluating flows through the heart valves, vascular structures and heart chambers [8]; measuring velocity (U) data throughout the cardiac cycle in all three spatial directions; and enabling the visualization of multidirectional flow patterns [9]. Furthermore, the acquisition of a single volume allows the user flexibility for the retrospective placement of analysis planes at any point within that volume [10]. Measurements from 4D MRI have been validated, compared to 2D-PC MRI, as the primary method for the routine clinical quantification of aortic blood flow [11], overall showing good agreement in CHD [12,13,14,15,16,17,18]. Given the complex flow patterns typical of CHD [16,19], 4D MRI is emerging as a particularly appealing tool for hemodynamic assessment in CHD. In light of the opportunity of extracting U and vessel area (A) data at multiple locations along a volume, this also represents an opportunity to derive additional hemodynamic information at the post-processing stage by means of performing wave intensity analysis. This method allows for the assessment of the working condition of the heart in relation to the remainder of the vasculature (i.e., the ventriculo-arterial coupling). Wave intensity was introduced over 30 years ago for visualising and quantifying arterial waves [20]. Wave intensity, traditionally defined as the product of pressure and velocity differentials over a time interval [21], can also be derived from PC-MRI [22] and has been applied to CHD scenarios, providing valuable hemodynamic insights. Here, we wanted to assess the feasibility of performing the analysis from 4D MRI data, thus potentially extending its applicability and representing a promising tool for future studies on arterial wave dynamics.

## 2. Materials and Methods

### 2.1. Study Participants

Patients scheduled for aortic replacement (e.g., elephant trunk/whole arch replacement, Ross procedure and Ozaki procedure) were prospectively identified based on their BAV morphology and invited for an MRI scan. Healthy individuals without any known cardiovascular disease were recruited as controls. A total of *n* = 12 BAV patients and *n* = 7 volunteers were recruited for the study. Exclusion criteria included patients with previous aortic valve replacement (AVR) or other aortic interventions, patients with connective tissue disorders or syndromes, patients with metal body implants or permanent metal fragments, pregnancy, claustrophobia and severe kidney dysfunction (i.e., estimated glomerular filtration rate < 60 mL/min/1.73 m^2^) [23]. Demographic and clinical data were obtained from the patients’ hospital records, including valve phenotype (i.e., right-left (RL), right-non-coronary (RNC) and left-non-coronary (LCN) fusion), degree of aortic stenosis (AS) and regurgitation (AR), type of dilation (one sided/overall) and presence of calcification. Imaging acquisition received Health Research Authority approval after being reviewed by the Research Ethics Committee (REC) (REC reference: 17/NI/0147).

### 2.2. Image Acquisition and Segmentation

The MRI scans were acquired at 3T (Magnetom, Skyra, Siemens Healthineers, Erlangen, Germany). An aortic 4D flow sequence was included for haemodynamic assessment. Sequence settings were as follows: velocity encoding (VENC): 150–400; temporal resolution (TR): 38.36 ms; echo time (TE): 2.1 ms. The acquisition time was 10–12 min.

Segmentation was performed with commercial software (CVI42 Flow package, Circle Cardiovascular Imaging, Calgary, Canada), which has been previously used for analysis of 4D flow data [24,25,26,27]. The appropriate volume (“mask”) was selected to encapsulate the whole aorta based on its anatomical markers. The aortic centreline was set automatically by the software and a manual threshold for the aortic volumes was set so that it would best fit the centreline at the highest flow through the vessel in systole. Upon reconstruction, the software requires position one or multiple planes perpendicular to the centreline along the vessel to perform analysis, including calculation of A and U waveforms at each plane. Three planes along the aorta were chosen for analysis: (1) in the ascending aorta above the sinotubular junction; (2) in the thoracic aorta just after the brachiocephalic vessels; (3) in the descending aorta above the diaphragm. Position 1 was chosen due to the anatomical (hence, haemodynamic) variability typical of BAV patients. Position 2 was selected as the brachiocephalic bifurcations in the aortic arch, which represents the first major (lumped) reflection site along the aorta. Position 3 was selected in order to establish any further changes as the waves travel into the descending aorta.

Flow streamlines, flow displacement in the aorta and flow rate were measured. To allow precise delineation of the aortic wall, the operator was allowed to alter regions of interest (ROI) at each plane, and careful vessel segmentation for each frame was performed manually. The adjusted ROI was then used to calculate A and pinpoint the location in the phase image, which was used to calculate the U signal. The extracted A and U signals were then used to calculate wave intensity, as briefly explained in the next paragraph.

### 2.3. Wave Intensity Analysis

The A and U signals were processed with an in-house script (Matlab, MathWorks, Natick, MA, USA). The extracted values were smoothened using a Savitsky–Golay filter (window size: 5–11), due to the heterogeneity of the 4D data, and a second-degree polynomial. Separation of forward and backward wave components requires the measurement of wave speed *c* [21,22]. The Bramwell–Hill equation relates *c* to arterial distensibility [21], where ρ is blood density, and D is distensibility, which equals relative area change (*dA*/*A*) divided by pressure change (*dP*):(1)$$c^{2}=\frac{1}{\rho D}=\frac{AdP}{\rho dA}$$

The water hammer equation also shows how *c* relates to *dP* and U change (*dU*), where the “+” subscript indicates forward waves and the “−” indicates backward ones: (2)$$dP_{\pm }=\pm \rho cdU_{\pm }$$

*c* using non-invasive parameters (i.e., replacing *dP* in Equation (1) with Equation (2)) can be expressed by the following equation:(3)$$c=\frac{AdU}{dA}=\frac{dU}{dlnA}$$

This can also be arranged to express the water hammer in terms of U and A:(4)$$dU_{\pm }=\pm cdlnA_{\pm }$$

Similarly to the PU loop method [28], U and lnA data were thus plotted to produce a U-lnA loop (Figure 1). The early systolic linear part of the loop yields *c* [29].

Considering arterial waves as a product of the summation of incremental wave fronts [21], U and A curves can be separated into their forward and backward constituents:(5)$$dU=dU_{+}+dU_{-}$$
(6)$$dlnA=dlnA_{+}+dlnA_{-}$$

These can be rearranged to obtain the following equations:(7)$$dU_{\pm }=\frac{1}{2}\left(dU\pm cdlnA\right)$$
(8)$$dlnA_{\pm }=\frac{1}{2}\left(dlnA\pm \frac{1}{c}dU\right)$$

The separated area-based intensity *dI_A_* can be derived as product of *dU_±_* and *dlnA*_±_ above, whereas net *dI_A_* is directly calculated as the product of the U and lnA differentials:(9)$$dI_{A}=dUdlnA$$

The resulting wave intensity pattern was analysed, at each plane, extracting peaks (i.e., intensity) and areas (i.e., energy) of the three typical dominant waves in systole, which include a forward compression wave (FCW) as a surrogate of left ventricular contractility, a forward expansion wave (FEW) as a surrogate of isovolumic relaxation and a backward compression wave (BCW) describing wave reflection. We refer the reader to the broader wave intensity literature for additional details on the methodology and clinical significance.

### 2.4. Statistical Analysis

The analysis was performed in Stata (v.13.1, StataCorp, College Station, TX, USA). Continuous data are reported as mean ± standard deviation (SD) and categorical variables as percentages. The chi-square test was used to compare proportions. Differences between two continuous variables were tested with a Mann–Whitney test (e.g., BAV vs. control), and a Kruskal–Wallis test was performed to test continuous variables against independent ones, followed by a Dunn’s test post hoc (e.g., position 1 vs. position 2 vs. position 3). Univariate linear regression analysis was used to test for associations between clinical and anatomical variables as appropriate. Alpha level was set at *p* < 0.05.

## 3. Results

Out of the 19 participants who entered the study, 17 were eligible for analysis. One healthy volunteer and one BAV patient were excluded due to sub-optimal image quality. Patients (*n* = 11) were 54 ± 11 years old and 50% male with a BSA of 2.0 ± 0.2 m^2^, while healthy volunteers (*n* = 6) were 41 ± 14 years old and 54% male with a BSA of 2.0 ± 0.07 m^2^. Details about dilation type, valve phenotype, the degree of AS and/or AR and the presence of calcification were not available for one BAV patient; nonetheless these characteristics (which were not used for any analyses) are reported for completeness in Table 1.

The aortas of all participants were successfully reconstructed (Figure 2). The visual assessment of the flow velocity and U streamlines of the two groups showed high U with turbulent helical flow in the BAV group and a lower, more laminar profile in the healthy volunteers, as expected. The successful extraction of U and A during the cardiac cycle for all participants was also performed from all three planes (the waveforms are shown in Figure 3).

Examining the physiological data revealed a significantly higher mean U (U_mean_) (*p* = 0.004), maximum systolic A (A_max_) (*p* = 0.0009) and minimum diastolic A (A_min_) (*p* = 0.0009) in the ascending aorta of the BAV population compared to the controls (Table 2).

Overall, there was no significant difference in peak U between the planes in the BAV patients (*p* = 0.19) and the control population (*p* = 0.67). Similarly, no overall plane differences in mean U were seen in the BAV patients (*p* = 0.15) and the control group (*p* = 0.36).

The A_max_ during systole was significantly different across the three planes in the BAV group (*p* = 0.0001) and in the control group (*p* = 0.0016), and, similarly, the A_min_ during diastole was also significantly different across the three planes in the BAV group (*p* = 0.0001) and the control group (*p* = 0.0005).

An example of wave intensity signals generated in a BAV patient and a healthy volunteer are shown in Figure 4. Throughout all measurement sites, the control group exhibited higher c compared to the BAV group (Table 3). Overall, there was no significant difference in *c* between the planes in the BAV population (*p* = 0.14) and the healthy volunteers (*p* = 0.34). The only significant difference was observed within the BAV group, between the thoracic aorta and the descending aorta (*p* = 0.03). No significant relationship was observed between age and *c* in the ascending aorta (*R*^2^ = 0.20, *p* = 0.067).

## 4. Discussion

This study examined the feasibility of performing wave intensity analysis based on area and velocity data derived from 4D MRI scans in patients with BAVs and healthy volunteers, assessing the physiologically realistic nature of the results. The analysis could indeed be performed successfully in all cases, except in two in which images were deemed to be of sub-optimal quality for a 4D flow analysis. The results were physiologically meaningful, with mean U and A changes during systole and diastole significantly higher in the ascending and thoracic aorta of the BAV population compared to the control group (see Table 2). No difference was observed in wave speed or any of the wave peaks. This study adds to the growing literature on wave intensity, demonstrating that the analysis can be performed using 4D MRI, hence allowing for the study of wave propagation over time in the same vessel during the same acquisition, which can be particularly insightful in CHD.

As illustrated in Figure 3, the U signals were realistic at all locations along the aorta, and a significantly higher mean U was observed in the ascending aorta of BAV patients. This is in line with previous literature findings in which U was also increased, which is largely due to the morphological nature of BAVs that leads to an eccentric flow jet [30,31,32]. This was also observed in the thoracic aorta. Although this comparison was not the main goal of the study and was limited by the sample size, the observation was intriguing, as flow abnormalities in BAVs have been reported to normalise at this point [33].

The observed significant difference in aortic area change is also physiologically sensible, as ascending aortic dilatation was present in 10 of the BAV patients (Table 1). As for the significantly higher A_max_ in the thoracic aortas of the BAV patients, this could potentially indicate that the thoracic aorta is able to accommodate for the increased U and, thus, reflect a slightly increased distensibility.

The calculated *c*, at all three planes within our sample, fit well with previously published studies. These studies have focused both on various cardiovascular diseases [34] and healthy individuals with varying age [35,36] using PC-MRI, which is considered the gold standard for measuring blood flow in the aorta. Our *c* measurements are also supported by a study that focused solely on BAV patients and reported an average *c* of 4.2 m/s (range: 2.5–6.3 m/s) [37]. As reported in Table 3, *c* differences were not significant between the two groups within the ascending and thoracic aorta. In the context of BAVs, a decreased aortic distensibility compared to controls has been reported [38,39]. Our study did not find a significant difference in *c* in the ascending and thoracic segments, suggesting such a change in distensibility. Firstly, this could mean that the flow-induced tissue changes observed in BAV patients might not occur in conjunction with the global increase in aortic stiffness, at least not to the extent to which it can be captured with *c* quantification from 4D MRI [40]. Secondly, the presence of valvular conditions, such as AS or AR, can affect *c* measurements [41]. Additionally, one echocardiographic study, which used the aortic stiffness index, found no significant distensibility changes between BAV patients with AS and controls and suggested that the presence of AS could progressively reduce distensibility (and, hence, increase *c*) [42]. The varying degrees and presence of valvular disease in our group, combined with the small sample size, likely explain these results.

Interestingly, the control group exhibited a significantly higher *c* in the descending aorta (*p* = 0.04) than the BAV patients. Previous studies have shown that BAV populations present with higher *c* in the descending aorta even in the absence of concomitant valvular disease [43]. One study, however, did find that BAV patients had greater distensibility than those with tricuspid aortic valves [41]. The homogeneity of the valvular disease in our population, as mentioned above, could be a possible explanation. It is possible to speculate a more distensible or enlarged descending aorta in BAV patients, which has also been observed using computational modelling [44], making that segment more compliant, which would lead to lower *c*, as the A_max_ in the BAV group was significantly higher compared to the controls (Table 2). This notion is also supported by the reduced aortic stiffness in patients with abdominal aortic aneurysms and the subsequent increase in stiffness after repair [45,46].

Numerous studies using different methodologies have concluded that *c* and, hence, stiffening increase linearly with age [47,48,49]. Our study did not find such a significant association (*p* = 0.067), which might be due to the small size of our population and its relatively youth (mean age 53.7 ± 11.4) compared to other studies, which had more varying age groups, some consisting of patients above the age of 66 [35]. Furthermore, there was no significant difference between the three main waves across groups, potentially suggesting that there is no major difference in VA coupling between the two groups. However, these observations should only be treated as preliminary and hypothesis-generating.

From a methodological standpoint, it is important to mention the temporal resolution at which the images were acquired. Lower temporal resolution is a known limitation of 4D MRI compared to the gold standard PC-MRI [8]. This could have affected *c* calculations as it is a parameter for which calculation requires high temporal resolution [22]. Future work in this context could experiment with higher temporal resolution 4D MRI.

From a clinical standpoint, the MRI scans used in this study were research and not clinical scans. While its value is increasingly recognised for clinical use, including in congenital heart disease and in aortic diseases [50], 4D flow is not routinely acquired as part of a cardiac MRI scan in many centres. A main limitation of 4D flow acquisition historically has been the long acquisition time, but techniques are increasingly being explored to reduce this and, thus, facilitate clinical implementation [51]. Nonetheless, demonstrating the feasibility of the technique opens the door to research applications and to the future exploration of clinical applications. Some examples of the clinical value of wave intensity include its potential to assess sub-clinical ventricular dysfunction [52] or to stratify hypertensive patients with different ventricular configurations [53]. The clinical relevance of wave intensity includes both mechanistic and prognostic insights, and it has been shown to be a highly sensitive measure in different scenarios, from aortic stenosis to left ventricular hypertrophy and heart failure [54]. Wave analysis has the potential to complement MRI-derived biomechanical and hemodynamic information with data on wave travel, wave reflection and ventriculo-arterial coupling.

## 5. Limitations

The study has several limitations. The first one is the small sample size, which inherently limited our ability to compare the BAV patients and the controls. Nevertheless, the aim of this work was to assess the feasibility of the proposed methodology to obtain realistic physiological data, which was achieved. This, in turn, would allow future research to generate appropriate hypotheses on wave parameters and analyse them accordingly, which will be interesting across several CHD scenarios.

The heterogeneity of the BAV population represented an additional limitation. The group had varying degrees and presences of valvular disease as well as different phenotypes of BAVs (Table 1). This could likely lead to an increase in the variability of the results as the different pathoanatomical features of BAVs can correspond to different unfavourable haemodynamic environments [55].

The segmentation method of this study is also a methodological point that warrants discussion as it still requires a significant amount of manual input. Although measuring A presents the advantage of not assuming vessel circularity [22], which can be particularly relevant in some CHD scenarios, e.g., after surgical repair, it is interesting to potentially move away from manual segmentation, strengthening the potential for future clinical applications of this methodology and similar ones. Current technological improvements have explored the use of machine learning to speed up the acquisition, flow measurement and—importantly—the analysis process with MRI data [56]. This would be particularly useful if the discussed methodology is applied to large cohorts of patients as 4D MRI datasets are inherently of a large volume [56].

Future work should apply the methodology to larger samples of patients and volunteers. This would not only improve the robustness of the results but also allow more in-depth analyses that explore variables including the BAV morphology and the additional assessment of haemodynamic parameters to compare against the wave intensity data. Increasing the temporal resolution would allow the easier manual handling of the segmentation as well as more reliable calculations of *c*.

## 6. Conclusions

This work demonstrates that it is feasible to derive wave intensity analysis from 4D MRI in patients with bicuspid aortic valve disease and in healthy volunteers. Our findings are in line with the physiological changes during the cardiac cycle and are supported by validated methodologies, including physiological values of wave speed, supporting the feasibility of the approach. Provided testing in larger cohorts, future research could employ our proposed methodology in order to assess clinical implementation.

## Figures and Tables

**Figure 1 bioengineering-10-00662-f001:**
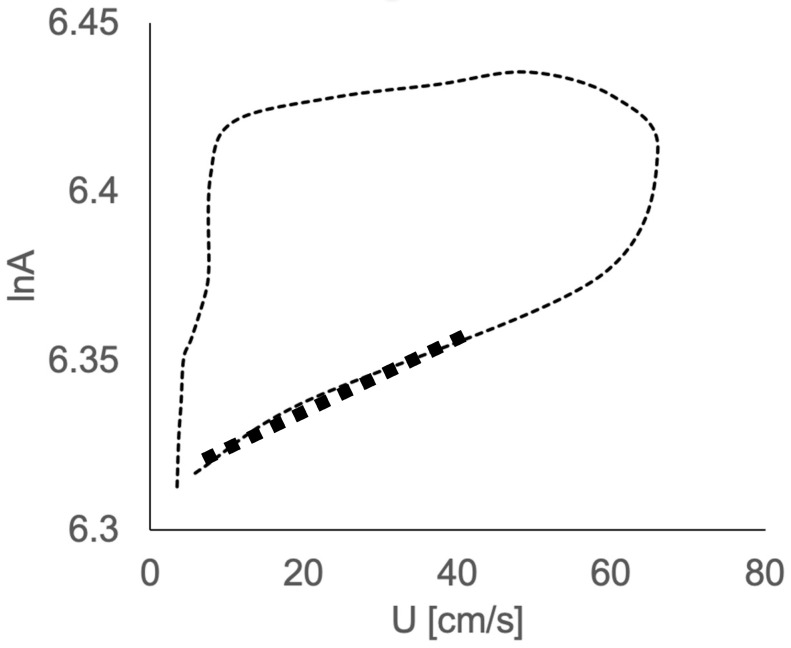
Example of velocity–area (U-lnA) loop, with the linear portion of the initial part of the loop yielding wave speed (in bold).

**Figure 2 bioengineering-10-00662-f002:**
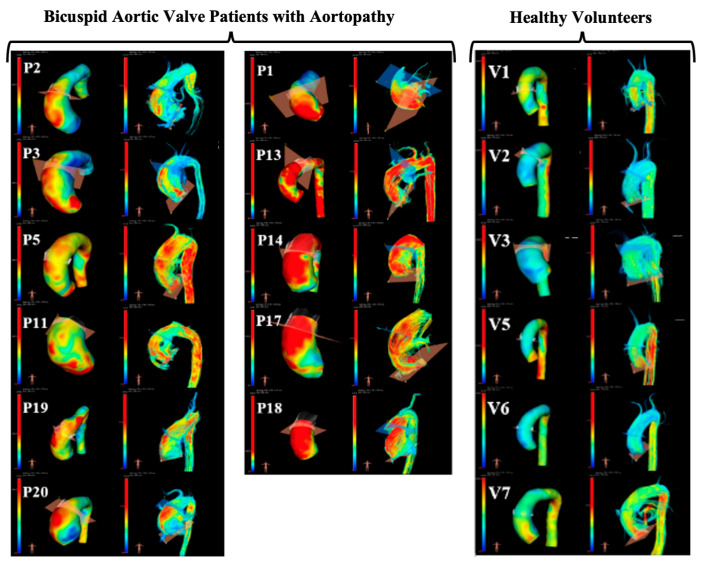
Illustration of 4D MRI analysis and qualitative comparison of wall shear stress maps and velocity streamlines (left and right in each pair of images, respectively). Patients (P) and volunteers (V) are all presented, and it is possible to qualitatively appreciate higher wall shear stress (yellow and red regions) and more helical flow in patients with bicuspid aortic valve, as expected.

**Figure 3 bioengineering-10-00662-f003:**
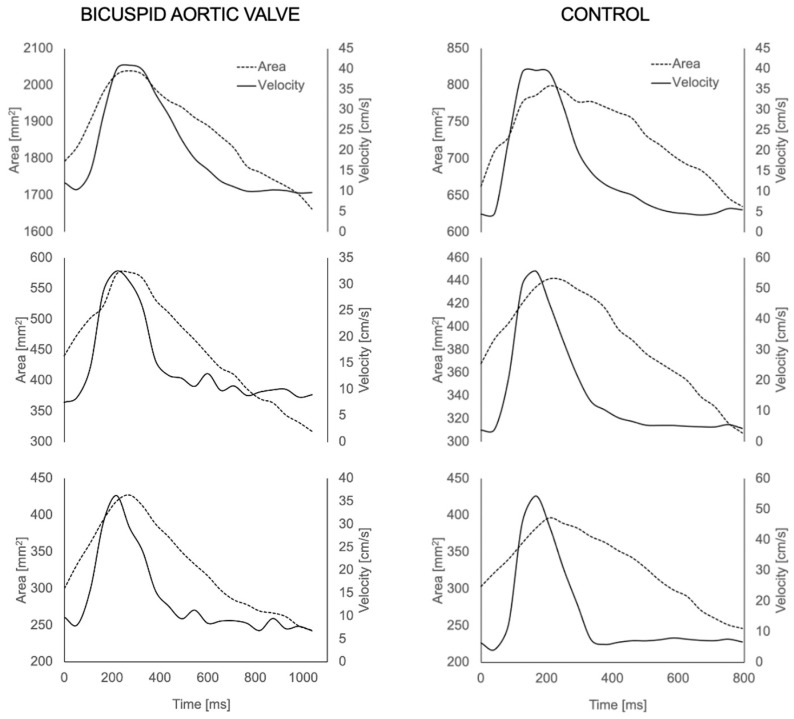
Area (A) and velocity (U) waveforms for each aortic plane (ascending, thoracic and descending aorta) extracted from the 4D MRI for a BAV patient (**left**) and healthy volunteer (**right**).

**Figure 4 bioengineering-10-00662-f004:**
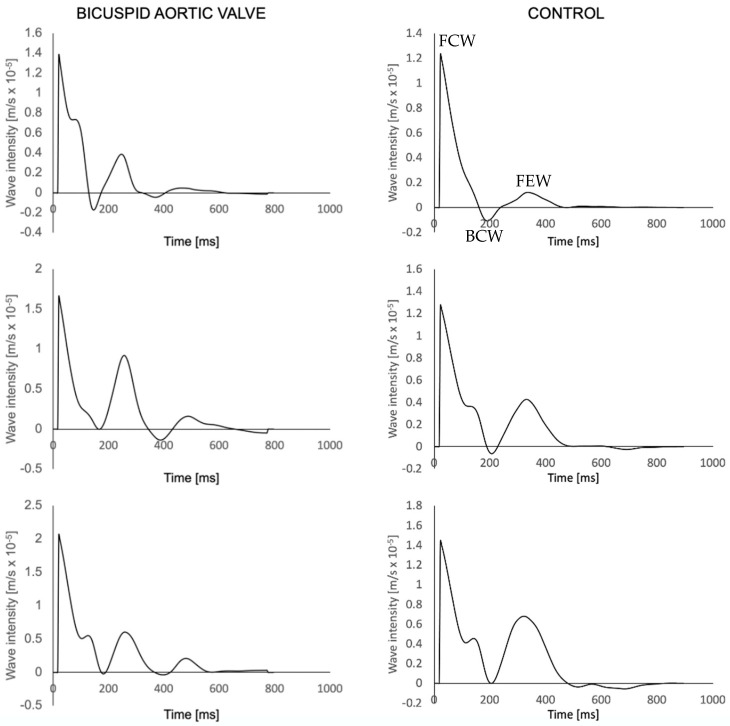
Wave intensity waveforms of a BAV patient (**left**) and a healthy volunteer (**right**) at the three aortic planes, sequentially from top to bottom. FCW = Forward compression wave; BCW = backwards compression wave; FEW = forward expansion wave (shown once to illustrate peaks).

**Table 1 bioengineering-10-00662-t001:** Baseline descriptive characteristics. BAV = bicuspid aortic valve; RL = right-left; RNC = right-non-coronary; LCN = left-non-coronary.

	Control (*n* = 6)	BAV (*n* = 10)
Ascending aorta dilatation type		
One-sided (*n*, %)	-	6 (60)
Overall (*n*, %)	-	4 (40)
Valve phenotype		
RL (*n*, %)	-	6 (60)
RNC (*n*, %)	-	3 (30)
LCN (*n*, %)	-	1 (10)
Degree of aortic regurgitation		
None (*n*, %)	-	4 (40)
Mild (*n*, %)	-	0
Moderate (*n*, %)	-	3 (30)
Severe (*n*, %)	-	3 (30)
Degree of aortic stenosis		
None (*n*, %)	-	3 (30)
Mild (*n*, %)	-	2 (20)
Moderate (*n*, %)	-	2 (20)
Severe (*n*, %)	-	3 (30)
Calcification present (*n*, %)	-	3 (30)

**Table 2 bioengineering-10-00662-t002:** Summary of velocity (U) and area (A) data derived from 4D MRI.

	Ascending Aorta			Thoracic Aorta			Descending Aorta		
	Control	BAV	*p* Value	Control	BAV	*p* Value	Control	BAV	*p* Value
U peak (cm/sec)	53.49 ± 14.5	58.42 ± 16.62	0.22	52.94 ± 17.60	58.40 ± 26.90	0.91	57.26 ± 21.5	48.35 ± 20.76	0.42
U mean (cm/sec)	17.53 ± 3.73	33.74 ± 19.81	0.004	18.15 ± 1.83	32.16 ± 19.61	0.03	19.86 ± 3.54	26.92 ± 14.83	0.68
A_max_ (mm^2^)	846.85 ± 197.2	1796.58 ±353.82	0.0009	496.90 ± 197.27	739.044 ± 192.45	0.0049	411.43 ± 41.59	602.16 ± 173.73	0.0049
A_min_ (mm^2^)	659.02 ± 115.82	1452.337 ± 303.52	0.0009	310.99 ± 17.13	437.43 ± 233.69	0.056	247.11 ± 34.47	322.94 ± 165.58	0.68

**Table 3 bioengineering-10-00662-t003:** Summary of wave intensity analysis results including wave peaks (FCW = forward compression wave; FEW = forward expansion wave; BCW = backward compression wave).

	Ascending Aorta			Thoracic Aorta			Descending Aorta		
	Control	BAV	*p* Value	Control	BAV	*p* Value	Control	BAV	*p* Value
Wave speed m s^−1^	6.40 ± 1.62	4.85 ± 1.75	0.09	5.90 ± 1.67	4.83 ± 1.44	0.19	5.11 ± 1.53	3.67 ± 1.77	0.04
FCW (m/s) × 10^−5^	1.01 ± 0.43	0.814 ± 0.58	0.22	1.18 ± 0.77	1.46 ± 0.81	0.36	1.62 ± 0.73	1.61 ± 0.64	0.48
BCW (m/s) × 10^−5^	−0.22 ± 0.20	−0.06 ± 0.07	0.10	−0.22 ± 0.23	−0.19 ± 0.36	0.48	−0.35 ± 0.33	−0.18 ± 0.34	0.08
FEW (m/s) × 10^−5^	0.08 ± 0.04	0.18 ± 0.11	0.07	0.27 ± 0.19	0.59 ± 0.58	0.22	0.47 ± 0.30	0.50 ± 0.55	0.54

## Data Availability

Data requests to be addressed to the corresponding author.

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
