# Peer review of "Feasibility of Wave Intensity Analysis from 4D Cardiovascular Magnetic Resonance Imaging Data"

_bioengineering, 2023, doi:10.3390/bioengineering10060662_

Round 1

Reviewer 1 Report

This is clearly written manuscript examining the use of 4D flow in patients undergoing aortic surgery. The number of patients is small, and as yet the findings relating velocity and area do not have a clear clinical correlate. However, the technique may be useful in the future, and is of interest to those who work in the area. Formatting and grammar are excellent.

Minor issue: U is defined in the body of the manuscript. I believe A is only defined in the abstract. I suspect it should also be defined early in the manuscript body.

Author Response

Reviewer 1

This is clearly written manuscript examining the use of 4D flow in patients undergoing aortic surgery. The number of patients is small, and as yet the findings relating velocity and area do not have a clear clinical correlate. However, the technique may be useful in the future, and is of interest to those who work in the area. Formatting and grammar are excellent.

We thank the Reviewer for their positive comments.

Minor issue: U is defined in the body of the manuscript. I believe A is only defined in the abstract. I suspect it should also be defined early in the manuscript body.

Both velocity and area are defined (as U and A, respectively) in the Introduction (page 3). 

Reviewer 2 Report

This study aims at characterizing the aortic hemodynamics in patients with bicuspid aortic valve in comparison with aortic hemodynamics in healthy control subjects using phase-contrast magnetic resonance imaging. Though the topic is interesting, it is unclear how the potential gathered information may be of use in the clinic.

Regarding the methods, how long is the 4D flow sequence? How feasible would it be to include it in a clinical MR scan?

Also, has the used software for segmentation (CVI42 Flow package, Circle Cardiovascular Imaging) being used in the past for this purpose? Please add reference to validate its accuracy and understand any simplifications/assumptions made by the software to solve for the stremalines and flow measurements.

A main hurdle in reading this manuscript has been the lack of coherence between the text and the figure captions. It has been rather confusing and has made it very difficult to get to the main message of the manuscript. Here you have a summary of this lack of coherence:

The first problem is that the figures appear out of order: page 4 has Figure 4, the first figure of the manuscript. The first Figure mentioned in the text, however, (page 5, line 178) is Figure 2. The labeld Figure 2 (included in page 6, lower half of the page) matches what is described in the text, but should not this Figure be labeled as Figure 1, though? Also, regarding the figure itself, the streamlines are barely visible. On the page that includes this Figure 2, there is a labeled Figure 1 on the upper half of the page, which should be placed on page 4, where the labeled Figure 4 appears.

The text mentions a Figure 5, but there is not any figure labeled as Figure 5. However, the labeled Figure 4 on page 4 corresponds to the description given in the main tex for Figure 5. At the same time, the text mentions a Figure 4, page 3 line 130, which does not correspond to the labeld Figure 4, but to the labeled Figure 1.

Table 2, mentioned in page 8 line 207, does not include any compliance measurement, or does Figure 5/4 also mentioned in that secion (page 7 line 206).

The Discussion mentions Table 3 on line 220, but it should be mentioning Table 2. Is the content of Table 3 mentioned in the text at all? 

In addition, there are some grammar errors, some examples:

Page 2 line 49 "compared 2D-PC MRI", shouldn't it be "compared to 2D-PC MRI"?

Page 2 line 52 "4D MRI is emerging a particularly appealing tool",  shouldn't it be "emerging as"?

Page 2 line 69 "MRI imaging" , shouldn't it be "MR imaging"?

These are some examples among others.

Author Response

This study aims at characterizing the aortic hemodynamics in patients with bicuspid aortic valve in comparison with aortic hemodynamics in healthy control subjects using phase-contrast magnetic resonance imaging. Though the topic is interesting, it is unclear how the potential gathered information may be of use in the clinic.

Regarding the methods, how long is the 4D flow sequence? How feasible would it be to include it in a clinical MR scan?

We thank the reviewer for this comment. We have reported the duration of the acquisition and have also included a note on the current clinical use of 4D MRI. We also note that, whilst the clinical use of 4D MRI is currently limited, the potential for research is also recognised and implementation of wave intensity could add to insights in clinical research studies. The purpose of the article was to assess the feasibility of the technique, which can have direct research applications.

Manuscript (p.10): From a clinical standpoint, the MRI scans used in this study were research and not clinical scans. Whilst its value is increasingly recognised for clinical use including in congenital heart disease and in aortic diseases[51], 4D flow is not routinely acquired as part of a cardiac MRI scan in many centres. A main limitation of 4D flow acquisition historically has been the long acquisition time, but techniques are increasingly being explored to reduce this and thus facilitate clinical implementation[52]. Nonetheless, demonstrating the feasibility of the technique opens the door to research applications and to future exploration of clinical applications.

Also, has the used software for segmentation (CVI42 Flow package, Circle Cardiovascular Imaging) being used in the past for this purpose? Please add reference to validate its accuracy and understand any simplifications/assumptions made by the software to solve for the stremalines and flow measurements.

We have provided additional references (refs. 25-28) as to the use of CVI42 for analysis of 4D flow MRI.

A main hurdle in reading this manuscript has been the lack of coherence between the text and the figure captions. It has been rather confusing and has made it very difficult to get to the main message of the manuscript. Here you have a summary of this lack of coherence: The first problem is that the figures appear out of order: page 4 has Figure 4, the first figure of the manuscript. The first Figure mentioned in the text, however, (page 5, line 178) is Figure 2. The labeld Figure 2 (included in page 6, lower half of the page) matches what is described in the text, but should not this Figure be labeled as Figure 1, though? Also, regarding the figure itself, the streamlines are barely visible. On the page that includes this Figure 2, there is a labeled Figure 1 on the upper half of the page, which should be placed on page 4, where the labeled Figure 4 appears.
The text mentions a Figure 5, but there is not any figure labeled as Figure 5. However, the labeled Figure 4 on page 4 corresponds to the description given in the main tex for Figure 5. At the same time, the text mentions a Figure 4, page 3 line 130, which does not correspond to the labeld Figure 4, but to the labeled Figure 1.

Table 2, mentioned in page 8 line 207, does not include any compliance measurement, or does Figure 5/4 also mentioned in that secion (page 7 line 206).

The Discussion mentions Table 3 on line 220, but it should be mentioning Table 2. Is the content of Table 3 mentioned in the text at all? 

We do apologise for the confusion in figure labelling. The article has been carefully checked now and all labelling has been amended.

In addition, there are some grammar errors, some examples:

Page 2 line 49 "compared 2D-PC MRI", shouldn't it be "compared to 2D-PC MRI"? Corrected

Page 2 line 52 "4D MRI is emerging a particularly appealing tool",  shouldn't it be "emerging as"? Corrected

Page 2 line 69 "MRI imaging" , shouldn't it be "MR imaging"? Corrected

These are some examples among others.

We have revised the whole manuscript and carefully checked grammar and spelling.

Reviewer 3 Report

Respectable authors demonstrated feasibility to derive wave intensity analysis from 4D MRI in patients with bicuspid aortic valve disease and in healthy volunteers. Since this is a brief report I have only minor comments.

1.  The authors need to clearly highlight what new information this manuscript provides.
2.  Also I would like to know the potential clinical value of wave intensity analysis from 4D MRI.
3. MRI  plays a unique role in these two sections: aortic biomechanical parameters and aortic flow hemodynamic parameters. Please comment if this wave analysis can add additional data on current practice.

Author Response

Respectable authors demonstrated feasibility to derive wave intensity analysis from 4D MRI in patients with bicuspid aortic valve disease and in healthy volunteers. Since this is a brief report I have only minor comments.

1.  The authors need to clearly highlight what new information this manuscript provides.

We have added a sentence in the abstract to highlight that this is the first study demonstrating the feasibility of deriving wave intensity from 4D MRI, which is the main novel information that the manuscript provides.

Manuscript (abstract): Demonstrating for the first time the feasibility of deriving wave intensity analysis from 4D flow data opens the door to research applications in different cardiovascular scenarios. 

Manuscript (p.8): This study adds to the growing literature on wave intensity, demonstrating that the analysis can be performed using 4D MRI, hence allowing to study wave propagation over time in the same vessel during the same acquisition, which can be particularly insightful in CHD.

  1.  Also I would like to know the potential clinical value of wave intensity analysis from 4D MRI.

We thank the author for this comment. The clinical application (and hence value) of clinical application of 4D MRI-derived wave intensity is interlinked with the clinical implementation of 4D MRI itself. This has been discussed in the revised manuscript. In addition, wave intensity has the potential of providing sub-clinical information in different scenarios, as well as detailed hemodynamic information. We have mentioned the clinical potential of wave intensity in the discussion.

Manuscript (p.10): Examples of the clinical value of wave intensity include its potential to assess sub-clinical ventricular dysfunction[53] or to stratify hypertensive patients with different ventricular configurations[54]. Clinical relevance of wave intensity includes both mechanistic and prognostic insights and it has been shown to be a highly sensitive measure in different scenarios, from aortic stenosis to left ventricular hypertrophy and heart failure[55].

  1. MRI  plays a unique role in these two sections: aortic biomechanical parameters and aortic flow hemodynamic parameters. Please comment if this wave analysis can add additional data on current practice.

Wave analysis has the potential to complement MRI-derived biomechanical and hemodynamic information with data on wave travel, wave reflection and ventriculo-arterial coupling. We have added this statement to the discussion.

Manuscript (p.10): Wave analysis has the potential to complement MRI-derived biomechanical and hemodynamic information with data on wave travel, wave reflection and ventriculo-arterial coupling.

Round 2

Reviewer 2 Report

I thank the authors for addressing the issues raised over the review process. I believe the study is better explained in the revised version.